# Full genome sequence analysis of African swine fever virus isolates from Cameroon

**Lynnette C. Goatley**[1�], **Graham Freimanis**[1�], **Chandana Tennakoon**[1�], **Thomas J. Foster**[1], **Mehnaz Quershi**[1], **Linda K. Dixon**[1], **Carrie Batten**[1], **Jan Hendrik Forth**[2], **Abel Wade**[3], **Christopher Netherton**[1]*

1 The Pirbright Institute, Ash Road, Pirbright, Woking, Surrey, United Kingdom, 2 Institute of Diagnostic Virology, Friedrich-Loeffler-Institut, Greifswald, Insel Riems, Germany, 3 National Veterinary Laboratory (LANAVET), Garoua, Cameroon

☌ These authors contributed equally to this work.
* Chris.netherton@pirbright.ac.uk

## Abstract

African swine fever (ASF) is a devastating disease of domestic pigs that has spread across the globe since its introduction into Georgia in 2007. The etiological agent is a large double-stranded DNA virus with a genome of 170 to 180 kb in length depending on the isolate. Much of the differences in genome length between isolates are due to variations in the copy number of five different multigene families that are encoded in repetitive regions that are towards the termini of the covalently closed ends of the genome. Molecular epidemiology of African swine fever virus (ASFV) is primarily based on Sanger sequencing of a few conserved and variable regions, but due to the stability of the dsDNA genome changes in the variable regions occur relatively slowly. Observations in Europe and Asia have shown that changes in other genetic loci can occur and that this could be useful in molecular tracking. ASFV has been circulating in Western Africa for at least forty years. It is therefore reasonable to assume that changes may have accumulated in regions of the genome other than the standard targets over the years. At present only one full genome sequence is available for an isolate from Western Africa, that of a highly virulent isolate collected from Benin during an outbreak in 1997. In Cameroon, ASFV was first reported in 1981 and outbreaks have been reported to the present day and is considered endemic. Here we report three full genome sequences from Cameroon isolates of 1982, 1994 and 2018 outbreaks and identify novel single nucleotide polymorphisms and insertion-deletions that may prove useful for molecular epidemiology studies in Western Africa and beyond.

## Introduction

African swine fever virus (ASFV) is a large double-stranded DNA virus that causes significant disease burden on domestic pigs and wildlife across the globe. Depending on the isolate the virus manifests an acute febrile illness that is invariably fatal, subacute disease in which a proportion of animals recover, or in some cases a chronic disease. Both domestic and wild pigs

**Data Availability Statement:** The assembled genomes generated in this study have been deposited in the NCBI GenBank database (www.ncbi.nlm.nih.gov) as OR387519, OR387520 and

OR387521 and the raw data are available in
BioProject PRJNA943480.

**Funding:** ASFV full genome sequencing at Pirbright
has been supported by UKRI grants BBS/E/I/
00007037, BBS/E/I/00007039, BBS/OS/GC/200015
and BBS/OS/GC/200015A and DEFRA grant
SE1517. This project has received funding from the
European Union's Horizon 2020 research and
innovation programme under grant agreement No
773701. The funders had no role in study design,
data collection and analysis, decision to publish, or
preparation of the manuscript.

**Competing interests:** The authors have declared
that no competing interests exist.

(*Sus spp.*) are fully susceptible. However, warthogs (*Phacochoerus africanus*) together with African soft ticks of the genus *Ornithodoros* represent a sylvatic cycle which acts as reservoirs for the virus in sub-Saharan Africa. Full genome sequencing of ASFV remains non-trivial due to difficulties in separating viral from host sequence and the presence of extensive homopolymers and repetitive regions that make *de novo* assembly problematic [1, 2]. Currently ASFV isolates are genetically typed based on a number of conserved and variable regions within the genome, with the genotype classified by 400 base pairs (bp) in the 3' end of the *B646L* gene that encodes for the major capsid protein p72 [3]. Genotypes can be further subdivided by differences in the *E183L* (p54 protein), *CP204L* (p30 phosphoprotein), [4, 5] as well as the intergenic sequence between the *I73R* and *I329L* genes [6]. The *B602L* gene (also referred to as the *9RL* gene) encodes for a chaperone required for the correct folding of p72 and contains a variable number of tetra amino acid repeats, referred to as the central variable region (CVR), that have proven useful in subtyping ASFV genomes in a number of studies [7, 8]. In addition, ASFV isolates can be categorised into serogroups based on cross-protection studies in pigs [9] and these serogroups map closely to the sequence of the *EP153R* (C-type lectin) and *EP402R* (CD2v protein) genes [10].

ASFV isolates from Western Africa are predominately *B646L* genotype I [11–14] with a single outbreak of genotype II in Nigeria in 2020 [15]. With the exception of some isolates from Senegal which are CD2v serogroup 1, these viruses are all CD2v serogroup 4 [12]. ASFV was first reported in Cameroon in 1981, with further outbreaks reported endemically in 1985 up to 2010 [13, 16], with the most recent in 2020 [17]. Isolates obtained from Cameroon can be subclassified from *B646L* genotype I and CD2v serotype 4 into *E183L* genotype Ia and Ib and variants containing 6, 19, 20, 21 and 23 tetra amino acid repeats in the pB602L protein [13, 17]. Temporal analysis suggests evolution of ASFV from strains containing 19 tetra amino acid repeats to 20 and then most recently 21 in the Far-North region of Cameroon. Virus containing 23 amino acid repeats were only reported in the initial outbreak in 1982 and have not been reported since then, while viruses containing 6 amino acid repeats have only been recently reported and appear to represent a new mutation [17].

Analysis of genotype II isolates obtained from Europe and Asia has identified additional single nucleotide polymorphisms (SNPs) and insertion-deletions (indels) which suggests that over time mutations can occur in regions of the genomes other than those typically analysed [6, 18–20]. The observed differences in the *B602L* and *E183L* genes could be due to the evolution of the virus in Cameroon over time [13, 17] or repeated reintroductions from neighbouring countries. Changes over time in positions other than those typically characterised by Sanger based sequencing have been observed in Sardinia [21], where genotype I ASFV has persisted in the field since 1980. In this study, we generated full genome sequences of three Cameroon isolates obtained in 1982, 1994 and 2018 as a starting point to understand the genetic diversity of ASFV in this country.

## Materials and methods

### Viruses

Cameroon 1982 isolate (CAM1982) has been described previously [22]. In brief it was passaged through a domestic pig, then through primary bone marrow culture and back through a domestic pig and bone marrow culture. Cameroon 1994/1 (CAM1994) was collected from a domestic pig in Tamrngang Puiying, Bamenda, in September 1994. CMR/lab1/2018 isolate (CMR2018) was collected from a domestic pig in 2018. All virus samples were passaged twice through porcine bone marrow cultures before sequencing.

## Sample preparation

Viruses were cultured on mononuclear bone marrow cells that were collected from the long bones of pigs four to six weeks old [23]. Mononuclear cells were purified from crude bone marrow cell preparations by density gradient centrifugation ($1000 \times g$, 20 min, RT). Cells were cultured in RPMI, GlutaMAX, HEPES supplemented with 10% foetal calf serum, 1 mM sodium pyruvate, 100 IU/mL penicillin, 100 μg/mL streptomycin and 100 ng/mL porcine colony-stimulating factor for 3–5 days before virus was added.

Viruses were cultured until 90 to 100% cytopathic effect was observed (typically 3 to 4 days), supernatants were clarified of cells and debris by low speed centrifugation ($1000 \times g$, 5 min, 4˚C) and virions were then concentrated by ultracentrifugation ($13600 \times g$, 90 min, 4˚C). The virus was treated with turbo DNase for 30 minutes at 37˚C and the enzyme was removed using DNase Inactivation reagent.

DNA extractions were carried out using MagAttract HMW DNA kit (Qiagen) and the resulting genomic DNA isothermally amplified using a REPLI-g kit (Qiagen). The average size and quality of the amplified DNA was checked using a genomic screentape on a TapeStation (Agilent). The DNA was quantified using the Qubit dsDNA BR assay kit (ThermoFisher) and the ASFV genomes in the samples were quantified by qPCR using primers for *B646L* [24], only samples with greater than or equal to $5.0 \times 10^6$ genome copies in 2 μl were taken forward for full genome sequencing.

## Genome sequencing

Illumina library preparation was performed using the DNA Prep kit (Illumina), with an input of approximately 500ng of DNA. The resulting libraries were quality checked using a Tapestation 4220 (Agilent) and quantified with Qubit prior to dilution and pooling. Samples were multiplexed and run on a MiSeq (Illumina) using a 600 cycle v3 reagent kit and flow cell.

To obtain longer reads, genomic DNA samples (between 2–4μg) were individually barcoded using the Nanopore Native barcoding genomic DNA kit (NBD104) and Genome by ligation kit (LSK109) as per manufacturers protocols. Samples were grouped in to 6–8 genomes and each group was run on a MinION (MIN-101b) using a 9.4.1 MinION flow cell for 16 hours. After this period, the flow cell was flushed out using the flow cell wash kit (EXP-WSH004), before additional groups were added for further 24-hour period. Raw reads are available in BioProject PRJNA943480.

## Assembly

The assembly was performed using an in-house pipeline. The sequencing data was first trimmed and adaptors removed using Trim Galore as a quality control step to reduce noise. The reads were then sub-sampled so that there is an even coverage of k-mers across the reads in the sample. These reads were assembled using SPADes 13.3.1 [25]. The reads were then mapped to the contigs produced in the assembly. The unmapped reads in this step were again sub-sampled to have an even k-mer coverage, and the contigs produced by the assembly were designated as trusted contigs. SPADes was run again using these reads and trusted contigs as the input. This iterative process of sub-sampling and assembly was carried out until the percentage of unmapped reads drops below 5% or the set of unmapped reads did not change from the previous iteration. The nanopore reads were passed as input data at each stage of the iteration. Illumina reads were mapped back to the final assembled genomes to identify SNPs and correct assembly errors using Geneious Prime (Biomatters, Inc.).

Assemblies were annotated using Genome Automated Transfer Utility [26] with Benin 1997/1 (AM712239) as a reference strain and other open reading frames (ORFs) identified

manually using the transcription map of the Badajoz 1971 Vero adapted strain (ASU18466) [27]. Annotations were edited and processed with Geneious Prime and GB2Sequin [28].

## Sanger sequencing

The central variable region (CVR) within the *B602L* gene was amplified by PCR and sequenced on an ABI-3730 using primers 9RL-F (5′-AATGCGCTCAGGATCTGTTAAATCGG) and 9RL-R (5′-TCTTCATGCTCAAAGTGCGTATACCT). The B407L locus was amplified using B407L-F (5′-GAGATGCCTCAGACTCTGCATATT) and B407L-R (5′-ATGACCCTGA ATTTTCGCTTGACT) and E199L with E199L-F (5′-CCACTGGAAGGCATCAAACGGTA) and E199L-R (5′-ATGTCTTGCATGCCAGTTTCCAC).

## Results and discussion

Two viruses isolated in Cameroon in 1982 and 1994 were selected from the Pirbright Institute ASFV reference collection for genome sequencing as well as a third isolate obtained from a recent outbreak in 2018. Samples from the three isolates were subject to both nanopore and Illumina sequencing. The final *de novo* assembled genomes were 182,927 bp for Cameroon 1982 (CAM1982) strain, 183,179 bp for Cameroon 1994/1 (CAM1994) and 181,952 bp for Cameroon 2018/lab1 (CMR2018) with minimum Illumina read depth of 44, 237 and 97 respectively (Table 1 and S1 Fig).

The sequences of the three isolates shared a similar genome structure to that of the Benin 1997/1 isolate, the only currently available genome sequence of a field isolate from West Africa. CAM1982, CAM1994 and CMR2018 were 98.81%, 99.90% and 99.78% identical to the Benin nucleotide sequence respectively. Comparison to the Badajoz 1971 Vero adapted strain (Ba71v), the only currently available full genome sequence that includes the complete sequence of the terminal inverted repeats (TIRs), suggested that approximately 2,000 base pairs are missing from the left end of all three of the assembled genomes. The differences in length of the Cameroon genomes were primarily due to assembly of TIR sequence at the right end of the genome, with CAM1982 having an additional 800 bp compared to the Benin 1997/1 sequence and CAM1994 having an additional 1600 bp. The open reading frames (ORFs) identified were practically identical to those seen in the Benin 1997/1 isolate, with 164 ORFs present in CAM1982 and CAM1994 and 162 in CMR2018. The extended assembly of the right-hand TIR in the CAM1982 and CAM1994 genomes allowed detection of orthologues of the *DP83L* and *DP93R* genes that are present in the Ba71v genome [29]. The Benin 97/1 genome was originally annotated with 156 ORFs, however orthologues of the *DP42R*, *C44L* and *J64R* genes as well as the novel genes *NG2*, *NG5* and *NG6* [27] are present in Benin 1997/1 as well all three Cameroon isolates.

Many of the differences among the three Cameroon isolates and Benin 1997/1 were single base pair insertions or deletions within intergenic homopolymers (S1 Table), however there

**Table 1. Assembly details for CAM1982, CAM1994/1 and CMR2018.**

| Samples (ASFV isolates) | Genome length (bp) | Proportion GC (%) | Nanopore depth (Min to Max) | Illumina depth (Min to Max) | Accession numbers |
|---|---|---|---|---|---|
| Cameroon 1982 (CAM1982) | 182,927 | 38.6 | 1–93 | 44–8,181 | OR387519 |
| Cameroon 1994/1 (CAM1994) | 183,179 | 38.5 | 3–184 | 237–9,854 | OR387520 |
| Cameroon 2018/lab1 (CMR2018) | 181,952 | 38.6 | 10–183 | 97–10,264 | OR387521 |

Final assembled genome length, the percentage of GC residues, the Nanopore and Illumina coverage across each assembled genome and the NCBI accession number are indicated.

are many homopolymers within genes and changes with some of these led to changes in the amino acid sequences. The *I196L*, *MGF360-16R* and *MGF360-18R* (*DP148R*) genes were truncated by 19, 48 and 59 aa respectively in the CMR2018 isolate, but these genes were not truncated in the CAM1982, or CAM1994 sequences when compared to Benin 1997/1. All three Cameroon isolates had an insertion at position 181868 that reconstituted the *DP60R* gene that was truncated in Benin 1997/1.

Differences in ASFV genome length are often due to variations in copy numbers of the five different multigene families (MGF) encoded by the virus, however with the exception of *MGF360-16R* and *-18R* the majority of the genes were no different to those seen in Benin. A number of SNPs were observed in different MGF360 and -505 genes and MGF360-2L in CMR2018 was truncated by 8 amino acids at the C-terminus. Changes in the predicted N-terminus of MGF110-11L in all three Cameroon genomes relative to Benin and in the N-terminus of MGF110-13L were detected, but these changes are not predicted to affect the presence of signal peptides or transmembrane domains.

The CMR2018/lab1 isolate contained 19 copies of the tetra amino acid repeats in the CVR (Table 2) and these were identical to isolates obtained from across Cameroon between 2010 to 2020, corresponding to Variant A in a previous study [13]. CAM1994/1 contained 28 tetra amino acid repeats that was mostly closely related to sequences obtained from Nigeria with 27 tetra amino acid repeats in 2003, 2004, 2015 and 2018 and with 29 repeats that was obtained in 2006 [11, 14, 30]. The assembled sequence of the CAM1982 CVR was identical to sequence AAQ08102 submitted to NCBI in 2003, but different to CAJ90777 [7] and inspection of the reads aligned to the final assembled 1982 genome suggested a mixed population in the sample. Inspection of the rest of the CAM1982 reads suggested two variants with the *E199L gene* (S1 Table). Therefore, individual clones of CAM1982 were generated by limit dilution and the sequence of the CVR, E199L and B407L loci determined by Sanger sequencing. As expected, all clones of CAM1982 had the novel deletion in B407L as predicted by the reads, however all eight possible versions of the B602L and the two E199L variants were present in the original sample. These differences have been annotated as polymorphisms in the final assembled CAM1982 sequence.

Indels of repetitive sequences that are present throughout the ASFV genome led to other differences between the Cameroon isolates and Benin 1997/1. CAM1982 has a single copy of ATGTTATAACC within the MGF360-9L/MGF360-10L intergenic region, whereas the other viruses have two. Differences in repetitive sequences within the *C44L* gene lead to 3 additional copies of ASTC within the protein sequence of the CMR2018/lab1 and one copy within the CAM1994/1 protein. An additional copy of a TCTTCACATTCA sequence within the *I215L*

**Table 2. B602L/CVR sequences from selected ASFV isolates.** Each letter in the CVR sequence represents a tetra amino acid repeat where A = CAST, B = CADT, C = GAST, D = CASM, F = CANT, N = NVDT, T = NVNT.

| Sample | CVR Code | Accession Number | Reference |
|---|---|---|---|
| Benin 1997/1 | ABNAAAACBNAAAAACBNAAAAACBNAAAACBNAFA | AM712239 | [31] |
| Cameroon 1982 | ABNAAAA(A)CBNABTDBNAFA | OR387519 | This study |
| Cameroon 1982 | ABNAAAACBNABTDBNAFA | AAQ08102 | |
| Cameroon 1982 | ABNAAAACBNABTDBNAAAAANA | CAJ90777 | [7] |
| Cameroon 1994/1 | ABNAAAACBNAAAAACBNAAAACBNAFA | OR387520 | This study |
| Cameroon 2018/lab1 | ABNAAAACBNABTDBNAFA | OR387521 | This study |
| Nig27-LGTT3_15 (Nigeria-Tet-27) | ABNAAAACBNAAAACBNAAAACBNAFA | KT961373 | [11] |
| CVR-Tet-27 | ABNAAAACBNAAAACBNAAAACBNAFA | GQ427187 | [30] |
| CVR-Tet-29 | ABNAAAAACBNAAAACBNAAAAACBNAFA | GQ427188 | [30] |

gene that encodes for the E2 ubiquitin ligase is present in the CMR2018/lab1 genome which leads to an additional DECE repeat in the amino acid sequence of the protein. All three viruses have a GCTTTGGACCGGCCG deletion within the B169L gene that leads to the deletion of one copy of three PAGPK repeats within the protein. There are also deletions within the *B407L* gene which lead to the deletion of one of the three copies of an NGSIR repeat from the CMR2018/lab1 protein and the deletion of two of three copies of NGSIR repeat and one of two copies of a SGSIR repeat in the CAM1982 protein, the CAM1994/1 *B407L* gene is identical to that of Benin 1997/1. Until full genome sequencing becomes routine a wider analysis of the copy number of these short repetitive sequences within ASFV isolated from Cameroon and West Africa could aid molecular epidemiological studies in the region.

Experimental studies with CAM1982 demonstrated a high virulence in pigs directly infected with high doses, but a lower virulence after transmission to contact pigs where a case fatality rate of 33% was observed [22]. Lower virulence was also reported after oral-nasal challenge with isolates obtained from Brazil and the Dominican Republic in 1978 [32, 33], although experiments with the genotype I of OUR T88/1 isolate demonstrated a high virulence in both directly infected and in contact pigs [34]. The principal differences between the CAM1982 genome and those of other highly virulent genotype I viruses such as Benin 1997/1 are the differences in the CVR and B407L. The influence of the CVR on the role of B602L in the correct assembly of p72 and viral virulence in swine is unknown, however the 26544/OG10 isolate of ASFV from Sardinia has a shorter CVRs than the Cameroon isolates and 26544/OG10 is virulent and replicates efficiently in macrophages [35]. *B407L* encodes for a late gene with unknown function that is expressed in infected macrophages [36, 37], but are not incorporated into virions [38]. All three of the Cameroon viruses have a small deletion in the *B169L* gene compared to Benin 97/1, however this deletion is present in a number of other genotype I viruses obtained in Portugal, Spain, Sardinia and the Dominican Republic [1, 39–41].

## Conclusions

Full genome sequencing of three isolates obtained across a nearly forty-year period did not identify major changes in the genome, however differences in the copy number of repetitive sequences were identified suggesting that ASFV is evolving in the field in Cameroon. A number of these changes were in regions in which variation has not been previously reported and therefore could represent novel targets for characterizing ASFV within Cameroon.

## Supporting information

**S1 Fig. Coverage plots.** Raw reads from Illumina and Nanopore sequencing runs were mapped back against the assembled genomes of CAM1982 (A), CAM1994/1 (B) and CAM2018/lab1 (C) using Geneious Prime. Plots were displayed using the Integrative Genome Viewer.
(PDF)

**S1 Table. Differences between CAM1982, CAM1994/1 and CMR2018/lab1 genomes and the Benin 1997/1 reference.** The positions of substitutions (>), insertions (^) and deletions (v) relative to the Benin 1997/1 sequence (AM712239) are indicated along with the gene within which the substitution is found, as well as any potential functional consequences.
(PDF)

## Author Contributions

**Conceptualization:** Linda K. Dixon, Carrie Batten, Abel Wade, Christopher Netherton.

**Data curation:** Christopher Netherton.

**Formal analysis:** Christopher Netherton.

**Funding acquisition:** Linda K. Dixon, Carrie Batten, Christopher Netherton.

**Investigation:** Lynnette C. Goatley, Graham Freimanis, Chandana Tennakoon, Mehnaz Quershi.

**Methodology:** Lynnette C. Goatley, Graham Freimanis, Chandana Tennakoon, Thomas J. Foster, Mehnaz Quershi, Jan Hendrik Forth.

**Supervision:** Lynnette C. Goatley, Carrie Batten, Christopher Netherton.

**Validation:** Thomas J. Foster, Christopher Netherton.

**Writing – original draft:** Lynnette C. Goatley, Graham Freimanis, Chandana Tennakoon, Christopher Netherton.

**Writing – review & editing:** Linda K. Dixon, Abel Wade, Christopher Netherton.

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
