## [Decision Letter · Decision Letter 0]

18 Oct 2023

PONE-D-23-32155Full genome sequence analysis of African swine fever virus isolates from CameroonPLOS ONE

Dear Dr. Netherton,

Thank you for submitting your manuscript to PLOS ONE. After careful consideration, we feel that it has merit but does not fully meet PLOS ONE’s publication criteria as it currently stands. Therefore, we invite you to submit a revised version of the manuscript that addresses the points raised during the review process.

We look forward to receiving your revised manuscript.

Kind regards,

Grzegorz Woźniakowski, Full professor, PhD, ScD

Academic Editor

PLOS ONE

Journal Requirements:

3. Thank you for stating the following financial disclosure: "ASFV full genome sequencing at Pirbright has been supported by UKRI grants BBS/E/I/00007037, BBS/E/I/00007039, BBS/OS/GC/200015 and BBS/OS/GC/200015A and DEFRA grant SE1517. This project has received funding from the European Union’s Horizon 2020 research and innovation programme under grant agreement No 773701."

Reviewers' comments:

Reviewer's Responses to Questions

**Comments to the Author**

1. Is the manuscript technically sound, and do the data support the conclusions?

Reviewer #1: Yes

2. Has the statistical analysis been performed appropriately and rigorously? 

Reviewer #1: Yes

3. Have the authors made all data underlying the findings in their manuscript fully available?

Reviewer #1: Yes

4. Is the manuscript presented in an intelligible fashion and written in standard English?

Reviewer #1: Yes

5. Review Comments to the Author

Reviewer #1: The article "Full genome sequence analysis of African swine fever virus isolates from Cameroon" fulfils the gaps in the knowledge about ASFV sequence in West Africa. The article is important and necessary for the data about ASFV evolution (strains from the period about 40 years). Authors made a great work with sequencing the isolates.

I have notice only minor notes. In my opinion even i abstract after full name of the disease and virus should be written the abbreviations in brackets, authors did not do that, using once full name and once the abbreviation.

In addition in the Results and Discussion section there are some sentences which should be in Materials and methods section (lines 144-150).

The article should be published after minor revision.

Sincerely,

Reviewer

6. PLOS authors have the option to publish the peer review history of their article (what does this mean?). If published, this will include your full peer review and any attached files.

Reviewer #1: No

---

## [Author Response · Author response to Decision Letter 0]

20 Oct 2023

Thank you for the time taken to review our paper. I accept all of the reviewers comments and have changed the manuscript accordingly.

The missing abbreviations have been added to the abstract.

The first part of the results and discussion was intended as a preamble, but appreciate that as written it was very methodical. We have shortened this to “Samples from the three isolates were subject to both nanopore and Illumina sequencing.”.

---

## [Editor Report · Decision Letter 1]

7 Mar 2024

PONE-D-23-32155R1

Full genome sequence analysis of African swine fever virus isolates from Cameroon

PLOS ONE

Dear Dr. Netherton,

Thank you for submitting your manuscript to PLOS ONE. After careful consideration, we feel that it has merit but does not fully meet PLOS ONE’s publication criteria as it currently stands. Therefore, we invite you to submit a revised version of the manuscript that addresses the points raised during the review process.

The authors have responded to the reviewers questions. 

We look forward to receiving your revised manuscript.

Kind regards,

Grzegorz Woźniakowski, Full professor, PhD, ScD

Academic Editor

PLOS ONE
---

## [Editor Report · Acceptance letter]

12 Mar 2024

PONE-D-23-32155R1 

PLOS ONE

Dear Dr. Netherton, 

I'm pleased to inform you that your manuscript has been deemed suitable for publication in PLOS ONE. Congratulations! Your manuscript is now being handed over to our production team.

Kind regards, 

on behalf of

Dr. Douglas Gladue 

Academic Editor

PLOS ONE